# Factors associated with the referral of children with severe illnesses at primary care level in Ethiopia: a cross-sectional study

Habtamu Beyene ![ORCID],1,2 Dejene Hailu Kassa,2 Henok Tadele,3 Lars Persson ![ORCID],4,5 Atkure Defar ![ORCID],5,6 Della Berhanu ![ORCID]4,5

▶ Prepublication history and additional online supplemental material for this paper are available online. To view these files, please visit the journal online. To view these files, please visit the journal online (http://dx.doi.org/10.1136/bmjopen-2020-047640).

For numbered affiliations see end of article.

**Correspondence to**
Mr Habtamu Beyene;
habtamu3346@yahoo.com

## ABSTRACT

**Context and objective** Ethiopia's primary care has a weak referral system for sick children. We aimed to identify health post and child factors associated with referrals of sick children 0–59 months of age and evaluate the healthcare providers' adherence to referral guidelines.

**Design** A cross-sectional facility-based survey.

**Setting** This study included data from 165 health posts in 52 districts in four Ethiopian regions collected from December 2018 to February 2019. The data included interviews with health extension workers, assessment of health post preparedness, recording of global positioning system (GPS)-coordinates of the health post and the referral health centre, and reviewing registers of sick children treated during the last 3 months at the health posts. We analysed the association between the sick child's characteristics, health post preparedness and distance to the health centre with referral of sick children by multivariable logistic regressions.

**Outcome measure** Referral to the nearest health centre of sick young infants aged 0–59 days and sick children 2–59 months.

**Results** The health extension workers referred 39/229 (17%) of the sick young infants and 78/1123 (7%) of the older children to the next level of care. Only 18 (37%) sick young infants and 22 (50%) 2–59 months children that deserved urgent referral according to guidelines were referred. The leading causes of referral were possible serious bacterial infection and pneumonia. Those being classified as a severe disease were referred more frequently. The availability of basic amenities (adjusted OR, AOR=0.38, 95% CI 0.15 to 0.96), amoxicillin (AOR=0.41, 95% CI 0.19 to 0.88) and rapid diagnostic test (AOR=0.18, 95% CI 0.07 to 0.46) were associated with less referral in the older age group.

**Conclusion** Few children with severe illness were referred from health posts to health centres. Improving the health posts' medicine and diagnostic supplies may enhance adherence to referral guidelines and ultimately reduce child mortality.

## INTRODUCTION

Globally, nearly 5.3 million children below the age of 5 years died in 2018. Of these deaths, 47% occurred during the first month of life.

### Strengths and limitations of this study

► Based on secondary analysis of register data from 165 health posts in 52 districts of the four most populous regions of Ethiopia, this study reports primary care referral patterns and associated factors in rural areas.

► This study used objectively measured data based on the WHO health service availability and readiness assessment tool and information obtained from the surveyed health posts compared with the national guidelines on referral.

► Using a cross-sectional study design, we cannot prove any causal relationship between the health post readiness, child factors and referral of sick children observed in this study.

► As the study was mainly based on register reviews, incomplete documentation of some of the health facilities might lead to reporting bias.

► The study findings do not give a complete picture of the country's referral system as we limited the analysis to the primary healthcare level, and we did not collect data from parents of the referred children.

More than half of this mortality occurred in sub-Saharan Africa. Infectious diseases remained the leading causes of death.[1]

Ethiopia achieved a significant reduction in child mortality during the Millennium Development Goals era from 1990 to 2015.[2] However, under 5 and neonatal mortality rates remain high with 55 and 30 deaths per 1000 live births, respectively.[3] Lower respiratory tract infection, diarrhoeal diseases and neonatal causes (mainly preterm birth complications, asphyxia and neonatal sepsis), were the leading causes of death.[4]

The integrated case management of childhood illnesses and the community based newborn care programmes are, if delivered with quality, cost-effective in reducing under 5 deaths.[5][6] Referral is included as a critical element in these programmes.[7] In Ethiopia,

the integrated community case management programme was initiated in 2010.[8] This programme trained health extension workers to assess, classify, and treat sick children 2–59 months of age with pneumonia, diarrhoea, malnutrition and other common illnesses and refer severe cases to health centres according to guidelines.[9] In 2013, the Community-Based Newborn Care programme was initiated. Health extension workers were trained to provide pre-referral antibiotic treatment for young infants (0–59 days) with possible severe bacterial infection and refer those to a higher-level facility. If a referral was not possible, the health workers were instructed to provide a 7-day antibiotic treatment. These interventions have been implemented in Ethiopia within the existing Health Extension Programme.[6 10]

A referral is defined as 'a process in which a health worker at one level of the health system, having insufficient resources (drugs, equipment, skills) to manage a clinical condition, seeks the assistance of a better or differently resourced facility at the same or higher level to assist in or take over the management of the client's case'.[11] A well-functioning referral system is an essential part of the health system and an indicator of the quality of care provided at health facilities.[7] The referral pattern may reflect the healthcare workers' ability to identify signs of severe illness at an early stage.[12] For effective referral care, a prerequisite is also that the higher-level facility is ready to receive and treat these children appropriately.[7]

In a previous study, we showed that the Ethiopian health extension workers referred only a fraction of the severely sick young infants and under 5 children that deserved urgent referral to health centres. Generally, referral logistics were weak.[13] Other studies in low-income countries have shown that the distance to the referral site, the severity of the child's illnesses,[14] and the availability of referral logistics[15 16] may influence severely sick children's referral.

This study aimed to identify factors associated with the referral of severely ill children aged 0–59 months and healthcare providers' adherence to referral guidelines in Ethiopia's primary healthcare system. Specifically, we assessed the referral of sick children 0–59 months from health post to the next higher level of healthcare in relation to guidelines. We analysed factors associated with referral care, including child characteristics, health post preparedness and distance to the next-level health facility.

## METHODS
### Study design and settings
This study was based on data from the evaluation of Optimising the Health Extension Programme intervention. This intervention was a collaborative effort between the Government of Ethiopia and implementing partners to increase the use and quality of community-based newborn care and integrated community case management of childhood illnesses in four Ethiopian regions.[17]

We did a secondary analysis of data from the end-line cross-sectional survey conducted in 26 districts, where the Optimising the Health Extension Programme intervention was implemented, and in 26 comparison districts without the intervention. These districts were found in four major Ethiopian regions, namely Amhara, Oromia, Southern Nations, Nationalities and Peoples and Tigray. The survey was conducted from December 2018 to February 2019, after the implementation of the intervention.

In Ethiopia, primary healthcare services are provided within the district health system, composed of five primary healthcare units and a district hospital serving around 100 000 people. Each primary healthcare unit consists of five health posts and their referral health centre. The health post is typically staffed with two health extension workers and serves a population of 5000 people.[18]

The referral care of sick children 0–59 months of age from health posts to the next level was assessed in relation to the treatment guidelines for 0–59 days old sick young infants (community-based newborn care), and 2–59 months old children (integrated community case management).

According to these guidelines, babies 0–59 days, who are very preterm (<32 weeks), very low birth weight (<1500 g), preterm (32–37 weeks), low birth weight (1500–2500 g), have a possible severe bacterial infection, severe jaundice, severe dehydration, severe persistent diarrhoea or dysentery require an urgent referral. Similarly, children 2–59 months with a possible severe bacterial infection, severe dehydration, severe pneumonia, possible severe bacterial infections, severe persistent diarrhoea, very severe febrile disease, severe complicated measles, mastoiditis, severe anaemia and complicated severe acute malnutrition require an urgent referral.[19]

### Study population and sample
The primary study had household and health facility modules to evaluate the Optimising the Health Extension Programme intervention.[20] We used the 2007 Ethiopian census to select 200 enumeration areas proportional to the population size of the 52 districts included in the study. In the main study, households were randomly selected to represent the enumeration areas. The health posts serving the selected households were included in this study. All records of 0–59 days old sick young infants, who sought care in 3 months prior to the survey, and records of the most recent ten 2–59 months sick children, who also sought care, were reviewed from treatment registers at the health post. Children without any illness who visited the health posts for postnatal care, vaccination, and growth monitoring were excluded from the analysis.

The sample size for the main study was based on the requirement that the baseline and end-line surveys should measure changes of a fixed number of percentage points of selected indicators for care-seeking and treatment for a sick child, based on household interviews.

## Data collection

The questionnaires used in this study were adapted from existing large-scale survey tools, such as the Demographic and Health Surveys and the Service Provision Assessment survey.[21] Questionnaires were translated into three local languages (Amharic, Afan Oromo and Tigrigna) and pretested. All data were collected using tablet computers.

Out of the eligible 200 clusters, 19 clusters were excluded from the study due to civil unrest. Trained health professionals and supervisors collected the data.

We collected data on the number of health extension workers at the health posts, availability of amenities, equipment, supplies, diagnostics and medicines. Geographical positioning system coordinates were assessed at health posts and referral health centres. We reviewed registers of 0–59 days and 2–59 months old children and collected background data on their symptoms, disease classifications, treatment, and referral status.

### Patient and public involvement

Not applicable (patients were not involved in this study).

## Data analysis

A descriptive analysis was performed, including frequencies, percentages and means. We characterised the health posts, the distance from health posts to the referral health centres, the proportion of health posts with service readiness tracer items and referral logistics for childhood referral care, and the referral of sick children of 0–59 months from health posts to higher levels in relation to the treatment guidelines.

We used health posts and their respective referral health centres' GPS readings to calculate the two's straight-line distance. The health post's facility readiness was assessed using the WHO service availability and readiness assessment (SARA) reference manual.[22] The general health facility service readiness has five domains, which include different tracer indicators, items and services. The first is basic amenities, including four items: communication, access to toilet facilities for clients, water source and power supply. The second is basic equipment with four items: infant or child scale, thermometer, functional stethoscope and mid-upper arm circumference tape measure. The third is standard precautions for infection prevention with seven items: sharps container, chlorine bleach, bucket for decontamination solution, contaminated waste container, soap and a towel or hand rub, alcohol-based hand rub and clean gloves. The fourth is the basic diagnostic capacity with a rapid diagnostic test for malaria. The last is essential medicines with ten items: vitamin A, tetracycline eye ointment, gentamycin, amoxicillin, oral rehydration solution, zinc-oral rehydration solution combined, zinc and the ready-to-use therapeutic foods (plumpy nut and BP100). Finally, the combined facility service readiness was assessed using the general service readiness index by calculating five domains' mean score.

The attributes of the referred sick children, their symptoms, disease classification and treatment were analysed based on the existing treatment guidelines.[19] Accordingly, the proportion of sick children with severe illnesses that require an urgent referral, non-severe illnesses that do not require urgent referral, the numbers referred and causes of their referrals were analysed. The referral status was linked with the facility readiness, and other health post factors, including the distance to the referral facility.

Referral of 0–59 days old sick young infants, and 2–59 months old children were the outcomes separately dichotomised as 'yes' (referred) or 'no' (not referred). $\chi^2$ tests were performed to assess whether the child's characteristics, health post preparedness and the distance to the next level health facility were associated with a sick child's referral. Relevant factors with statistically significant association to the referral of sick children in the bivariate analysis were entered into a multivariable logistic regression analysis to identify referral predictors. Adjustments were made for clustering and whether the cluster was an Optimising the Health Extension Programme intervention or comparison area. ORs with 95% CIs were calculated to measure the association between the predictor and outcome variables. Additionally, adherence to referral guidelines for sick young infants and older children was dichotomised as 'yes' (managed according to referral guideline) or 'no' (not managed according to referral guideline). Factors that had statistically significant association with referrals were then assessed in bivariate and multivariable logistic regression analyses to identify predictors of adherence to correct referral practices.

The Census and Survey Processing System software (United States Census Bureau, Washington DC, USA), was used to enter and edit survey data. Statistical analysis and tests were done using SPSS V.25 (SPSS, IBM).

## RESULTS

We extracted data from treatment registers at 165 health posts serving 181 enumeration areas. Information on the assessment, treatment and referral of 229 sick young infants of 0–59 days and 1123 sick children aged 2–59 months was collected from the registers. Out of the 165 health posts, 48 had registered to manage one or more 0–59 days old sick young infants, and 147 had registered one or more 2–59 months old sick children in the 3 months preceding the survey.

### Characteristics of health posts

Nearly 80% of health posts were staffed with two or more health extension workers, while the rest had only one. Almost all health posts had sick child registers, and 42% had referred at least one sick child to a health centre in the quarter before the survey. A cell phone signal was available in most of the surveyed health posts. Still, most health workers had neither used mobile phones nor other communication means for their most recent referral of a sick child (table 1). Half of the surveyed health posts were

**Table 1** Characteristics of health posts in four regions of Ethiopia (December 2018–February 2019) (n=165)

| Characteristics | n | % |
|---|---|---|
| Region | | |
| Amhara | 70 | 42 |
| Oromia | 48 | 29 |
| SNNPR | 22 | 13 |
| Tigray | 25 | 15 |
| No of health extension worker per health post | | |
| 1 | 35 | 21 |
| 2+ | 130 | 79 |
| Availability of | | |
| Registers for 0–59 days old children | 148 | 90 |
| Registers for 2–59 months old children | 156 | 95 |
| Cell phone signals in the health post | 135 | 82 |
| No of health posts that referred sick children in the last quarter | 69 | 42 |
| Means of communication used for the referral of last sick child | | |
| Facility landline or mobile phone | 3 | 2 |
| Staff member's mobile phone | 25 | 15 |
| In person communication | 13 | 8 |
| No communication for last child referral | 124 | 75 |

SNNPR, Southern Nations, Nationalities and Peoples Region.

found within 10 kilometres from the next level referral health centre (figure 1). There were missing distance data for seven health posts.

## Health post readiness status

The readiness of health posts to provide health services was assessed for five general service readiness domains. The average readiness score was 61% for basic amenities, 80% for basic equipment, 49% for standard precaution for infection prevention, 50% for basic diagnostic capacity, and 52% for essential medicines (figure 2). A significant proportion of the health posts lacked readiness to provide the required service for sick children, shown by an overall service readiness index of 57%. Moreover, nearly half of the health posts did not have the required essential medicines. Even amoxicillin, which is needed to treat possible severe bacterial infection of 0–59 days sick young infants, was not available in a quarter of the health posts. Rapid diagnostic tests, which are highly important in malaria endemic areas like Ethiopia, was not found in half of the health posts (figure 2). The items included in these scores are shown in online supplemental figures 1–4).

## Referral of sick children

During the last 3 months before the survey, 39 (17%) of the 229 examined 0–59 days old sick young infants were referred from the health posts to the nearest health centres. Similarly, 78 (7%) of the 1123 sick children aged 2–59 months old had been referred. Possible serious bacterial infection and pneumonia were the leading causes of referral among 0–59 days old and 2–59 months old children, respectively (table 2). Only 36% of 0–59 day sick young infants with possible severe bacterial infection were referred. Similarly, the health extension workers referred only 27% of 2–59 months old children with severe malnutrition (table 2). Cross-tabulation of the four facility readiness domains and their respective individual components are shown in online supplemental tables 1 and 2, respectively.

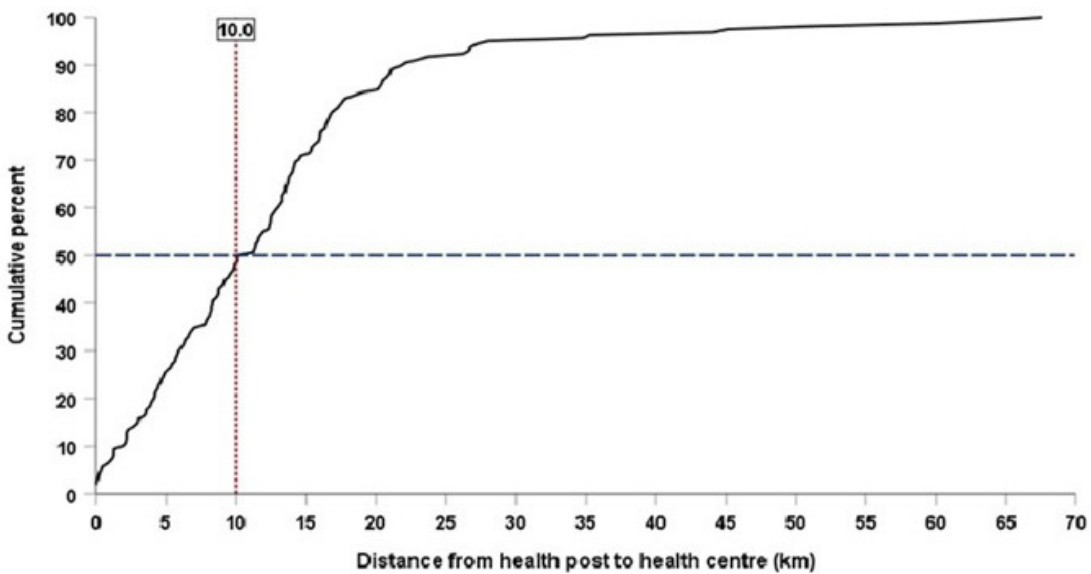

**Figure 1** Cumulative percentages of distance from health posts to the referral health centres, analysed based on linear distance from 158 health posts to their 128 referral health centres.

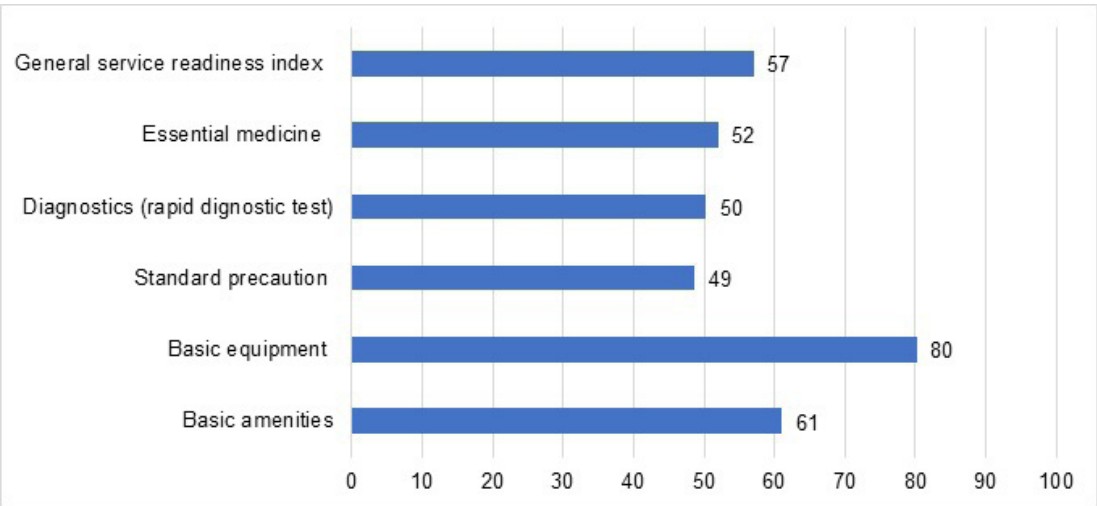

**Figure 2** The mean percentage scores for five general service readiness domains of the surveyed 165 health posts analysed based on the WHO Service Availability and Readiness Assessment reference manual.[22]

### Factors associated with referral

Sick young infants aged 0–59 days with severe illnesses were seven times more likely to be referred to the next referral level, as compared with those with non-severe illnesses (95% CI 1.50 to 30.11) (table 3). Sick young infants in the later weeks of life were less likely to be referred than those in the first week (adjusted OR, AOR=0.25, 95% CI 0.10 to 0.64). Neither the sex of the sick child nor the distance from health post to referral health centre were associated with referral (table 3). A higher proportion of sick young infants was referred from health posts served by two or more health extension workers than those served by only one (p=0.024) (online supplemental table 1).

Also, sick children aged 2–59 months with severe illness were more likely to be referred than those with non-severe illness (AOR=21.27, 95% CI 7.51 to 60.23) (table 3). Health posts that had a better score for basic amenities were less likely to refer 2–59 months sick children to the next level than those without (AOR=0.38, 95% CI 0.15 to 0.96). Health posts with amoxicillin (AOR=0.41, 95% CI 0.19 to 0.88) and rapid diagnostic tests (AOR=0.18, 95% CI 0.07 to 0.46) were less likely to refer 2–59 months old children than those without these medicines and diagnostic kit.

The number of health extension workers, age and sex of the child and distance from health post to the referral health centre were not associated with referrals of 2–59 months old children.

### Management according to guideline

The adherence to referral guidelines of sick young infants and older children is shown in table 4. Among the sick 229 young infants, 21 children were referred when a referral was not advised, and 31 children were treated at the health post when a referral was recommended. The remaining 177 (77%) were managed according to referral guidelines. Of the factors associated with a referral, essential medicines were also associated with adhering to referral guidelines for young infants. Those

in the higher tertile adhered more to referral guidelines (AOR=13, 95% CI 1.5 to 112.6). Among the 1123 sick older children, 56 were referred when a referral was not advised, and 22 children were treated at the health post when a referral was required. The remaining 945 (93%) were managed according to referral guidelines. Facilities with amoxicillin and diagnostics were more likely to adhere to referral guidelines (amoxicillin AOR=2.8, 95% CI 1.3 to 5.5 and diagnostics AOR=4.7, 95% CI 1.8 to 11.9). Data for this analysis are shown in tables 5 and 6.

### DISCUSSION

Our study from four major Ethiopian regions showed that health extension workers referred only around one-third of the sick young infants 0–59 days old and half of the 2–59 months old children that should be referred, according to guidelines. The leading diagnoses at referral were possible severe bacterial infections in young infants and pneumonia in older children. As expected, conditions classified as severe were referred more frequently than non-severe cases. The tendency to refer was associated with some characteristics. For 0–59 days old infants, young neonates were referred more often than a bit older young infants. Health posts with basic amenities, amoxicillin, and rapid diagnostic tests were less inclined to refer older sick children. Within the 2–59 months group, age was not associated with referral. There was no sex difference in referral. Further, the straight-line distance from the health post to the health centre was not associated with referrals.

Due to the cross-sectional study design, we cannot prove any causal relationship between the health post and child factors that were associated with a referral of sick children. Further, we have analysed referral from the first primary care level, and are therefore only displaying part of the country's referral system. Exclusion of records of few sick children due to incomplete documentation on registers of some of the health facilities might be a limitation

**Table 2** Disease classification, recommended and performed referrals among children 0–59 months old seen at the surveyed 165 health posts, December 2018–February 2019

| Age group | Disease classification | Referral guideline supports referral? | No seen | No (%) referred |
|---|---|---|---|---|
| 0–59 days | Very preterm or very low birth weight | Yes | 0 | 0 |
| | Preterm or low birth weight | Yes | 4 | 2 (50) |
| | Possible serious bacterial infection | Yes | 42 | 15 (36) |
| | Severe dehydration | Yes | 0 | 0 |
| | Severe persistent diarrhoea | Yes | 1 | 0 |
| | Dysentery | Yes | 1 | 1 (100) |
| | Severe jaundice | Yes | 1 | 0 |
| | Local bacterial infection | No | 59 | 5 (9) |
| | Some dehydration | No | 2 | 1 (50) |
| | No dehydration | No | 17 | 1 (6) |
| | Jaundice | No | 1 | 0 |
| | Malaria | No | 0 | 0 |
| | Feeding problem or low weight | No | 45 | 2 (4) |
| | Classification not given | – | 26 | 0 |
| | Other* | No | 30 | 12 (40) |
| | Total 0–59 days old | | 229 | 39 (17) |
| 2–59 months | Severe pneumonia/very severe disease | Yes | 9 | 8 (89) |
| | Severe dehydration | Yes | 2 | 1 (50) |
| | Severe persistent diarrhoea | Yes | 3 | 1 (33) |
| | Dysentery | Yes | 9 | 6 (67) |
| | Very severe febrile disease | Yes | 2 | 1 (50) |
| | Severe complicated measles | Yes | 0 | 0 |
| | Severe malnutrition | Yes | 22 | 6 (27) |
| | Severe anaemia | Yes | 0 | 0 |
| | Pneumonia | No | 478 | 16 (3) |
| | Some dehydration | No | 186 | 9 (5) |
| | No dehydration | No | 203 | 0 |
| | Persistent diarrhoea | No | 28 | 2 (7) |
| | Malaria | No | 12 | 0 |
| | Fever, malaria unlikely | No | 11 | 0 |
| | Fever, no malaria | No | 55 | 4 (7) |
| | Measles with eye/mouth complications | No | 0 | 0 |
| | Measles | No | 1 | 1 (100) |
| | Acute ear infection | No | 7 | 3 (43) |
| | Chronic ear infection | No | 1 | 0 |
| | Moderate malnutrition | No | 33 | 4 (12) |
| | Anaemia | No | 1 | 0 |
| | Others† | No | 64 | 16 (25) |
| | Total 2–59 months old | | 1123 | 78 (7) |

*Other—classification for 0–59 days included 'infection unlikely' (12), 'eye problems' (10) and post natal care (PNC) cases with referral outcome (8).
†Others—classification for 2–59 months old children included mainly 'no pneumonia, cough' (42), and 'eye problems' (6), growth monitoring with referral outcomes (16).

**Table 3** Factors associated with referral of 0–59 days old infants and 2–59 months old children from health post to higher levels in four Ethiopian regions, December 2018–February 2019

| Characteristic | Levels | Referred (n) | Not referred (n) | Crude OR (95% CI) | Adjusted OR (95% CI)* |
|---|---|---|---|---|---|
| **For 0–59 days old infants (N=228)†** | | | | | |
| The health post | | | | | |
| Facility owned landline or mobile phone | Not available | 13 | 35 | 1.0 | 1.0 |
| | Available | 26 | 154 | 0.47 (0.17 to 1.28) | 0.61 (0.13 to 2.80) |
| Mean score of essential medicine | 0%–33% | *1 | 11 | 1.0 | 1.0 |
| | 34%–66% | 32 | 112 | 3.14 (0.99 to 9.99) | 3.17 (0.42 to 24.19) |
| | 67%–100% | 6 | 66 | 1.00 (0.24 to 4.14) | 1.34 (0.10 to 18.80) |
| No of health extension workers | *1 | †1 | 39 | 1.0 | 1.0 |
| | 2 or more | 37 | 150 | 4.81 (1.10 to 21.05) | 5.63 (0.87 to 36.40) |
| Distance between health post and the nearest referral health centre | | | | | |
| Distance tertiles | 0–8 km | 14 | 60 | 1.0 | 1.0 |
| | 8.01–20 km | 11 | 63 | 0.54 (0.18 to 1.60) | 0.60 (0.18 to 2.04) |
| | 20.01 or more km | 14 | 66 | 0.87 (0.26 to 2.96) | 1.02 (0.33 to 3.17) |
| The child | | | | | |
| Age | 0–1 week | 14 | 45 | 1.0 | 1.0 |
| | 2–4 weeks | 16 | 82 | 0.63 (0.21 to 1.84) | 0.35 (0.11 to 1.10) |
| | 5 or more weeks | 9 | 62 | 0.47 (0.15 to 1.46) | 0.25 (0.10 to 0.64) |
| Gender | Boy | 18 | 105 | 1.0 | 1.0 |
| | Girl | 21 | 84 | 1.46 (0.59 to 3.62) | 1.12 (0.45 to 2.75) |
| Disease classification | Not severe | 21 | 158 | 1.0 | 1.0 |
| | Severe | 18 | 31 | 4.37 (1.30 to 14.67) | 6.72 (1.50 to 30.11) |
| **For 2–59 months old children (N=1074)‡** | | | | | |
| The health post | | | | | |
| Facility owned landline or mobile phone | Not available | 44 | 792 | 1.0 | 1.0 |
| | Available | 27 | 211 | 2.30 (1.16 to 4.55) | 1.88 (0.94 to 3.76) |
| Mean score of basic amenities | 0%–33% | 20 | 129 | 1.0 | 1.0 |
| | 34%–66% | 15 | 279 | 0.35 (0.13 to 0.89) | 0.38 (0.15 to 0.96) |
| | 67%–100% | 36 | 595 | 0.39 (0.17 to 0.89) | 0.40 (0.16 to 1.03) |
| Mean score of basic equipment | 0%–33% | †1 | 64 | 1.0 | 1.0 |
| | 34%–66% | 12 | 147 | 2.61 (0.65 to 10.45) | 3.49 (0.16 to 75.12) |
| | 67%–100% | 57 | 792 | 2.30 (0.62 to 8.53) | 2.38 (0.12 to 48.96) |

Continued

**Table 3** Continued

| Characteristic | Levels | Referred (n) | Not referred (n) | Crude OR (95% CI) | Adjusted OR (95% CI)* |
|---|---|---|---|---|---|
| Amoxicillin | Not available | 27 | 152 | 1.0 | 1.0 |
| | Available | 44 | 851 | 0.29 (0.15 to 0.58) | 0.41 (0.19 to 0.88) |
| Rapid diagnostic test for malaria | Not available | 45 | 449 | 1.0 | 1.0 |
| | Available | 26 | 554 | 0.47 (0.26 to 0.83) | 0.18 (0.07 to 0.46) |
| Mean score of overall service readiness‡ | 0%–33% | 6 | 22 | 1.0 | 1.0 |
| | 34%–66% | 43 | 573 | 0.28 (0.04 to 2.05) | 0.21 (0.03 to 1.53) |
| | 67%–100% | 22 | 408 | 0.20 (0.03 to 1.49) | 0.62 (0.07 to 5.60) |
| No of health extension workers | *1 | 20 | 196 | 1.0 | 1.0 |
| | 2 or more | 51 | 807 | 0.62 (0.30 to 1.28) | 1.01 (0.38 to 2.65) |
| Distance between health post and the nearest referral health centre | | | | | |
| Distance tertiles | 0–8 km | 24 | 321 | 1.0 | 1.0 |
| | 8.01–14 km | 25 | 320 | 1.18 (0.54 to 2.56) | 0.98 (0.43 to 2.21) |
| | 14.01 or more km | 22 | 362 | 0.86 (0.43 to 1.71) | 0.71 (0.28 to 1.79) |
| The child | | | | | |
| Age | 0–1 week | 25 | 284 | 1.0 | 1.0 |
| | 2–4 weeks | 24 | 317 | 0.86 (0.48 to 1.54) | 0.97 (0.52 to 1.81) |
| | 5 or more weeks | 22 | 402 | 0.62 (0.36 to 1.07) | 0.88 (0.45 to 1.68) |
| Gender | Boy | 35 | 526 | 1.0 | 1.0 |
| | Girl | 36 | 477 | 1.13 (0.71 to 1.81) | 0.89 (0.52 to 1.52) |
| Disease classification | Not severe | 50 | 981 | 1.0 | 1.0 |
| | Severe | 21 | 22 | 18.73 (9.21 to 38.09) | 21.27 (7.51 to 60.23) |

Logistic regression analyses.
*Multivariable logistic regression adjusted for region and clusters. The logistic regression model was tested using the Hosmer and Lemeshow test of the goodness-of-fit statistics, which suggested that the model was a good fit to the data.
†Records of 1 sick young infant of 0–59 days old were removed because of missing of data of distance from health post to the referral health centre.
‡Records of 49 sick older children of 2–59 months old were removed because of missing of data of distance from health post to the referral health centre.

**Table 4** Adherence to referral management guidelines for sick young infants and older children

| Age group | Observed | Management guideline | |
|---|---|---|---|
| | | **Referral advised** | **Referral not advised** |
| 0–59 days | Referred | 18 | 21 |
| | Not referred | 31 | 159 |
| 2–59 months | Referred | 22 | 56 |
| | Not referred | 22 | 1023 |

Note: Shadowed cells indicate referral management that did not adhere to guidelines.

which could lead to reporting bias. We used objectively measured data on the health posts, based on the WHO health SARA tool, and information obtained from the surveyed health posts were compared with national guidelines on referral. The study was performed in four of the largest regions in Ethiopia accounting for 80% of the population, where the enumeration areas were selected to represent the chosen districts of these regions. Though generalisation of the study findings to the regions might be difficult, it is expected that the selected districts did not differ in health systems characteristics.

Despite the WHO's recommendation,[23] our study showed that only a third of young infants and half of the older children, who required referral, were referred from health post to the health centres. This finding was consistent with the previous Ethiopian and Malawian studies.[9] [24] The low referrals seen might be attributed to health providers' fear that caregivers would protest a referral,[25] and the actual refusal of caregivers to comply with the referral.[26] These factors might lead the health extension workers to manage the child at the health post.[27] Healthcare providers might also undermine the overall impact of treatment guidelines on childhood survival.[25] [28] They may perceive that there is little added value in referral and even feel a loss of power and prestige when referring.[29] They might also lack the knowledge and be reluctant to adhere to the guidelines.[29] This low level of referral of severely sick children is concerning. It could delay access to appropriate care in the higher-level referral facilities, thus increasing the risk of death,[30] and hinder the efforts to reach the sustainable development goals on neonatal and under-five mortality.[31]

In this study, the severity of a child's illness was associated with their referral in both age groups.[32] This relation indicates that severely ill children were more frequently managed according to the referral guidelines. There is a need to strengthen health extension workers' ability to identify severely sick children requiring urgent referral.[33] Neonates were primarily referred in their first week of life, highlighting the need for urgent referrals of severely ill newborns to prevent death in neonatal sepsis and other critical conditions.[34] Also, we found that distance from health posts to health centre was not associated with referrals, perhaps since nearly three-quarters of the health posts were located within 15 kilometres from their referral health centre.

Lack of the overall readiness of the health posts to provide the required childcare services[22] is an important finding of this study. This scenario could indicate that

**Table 5** Factors associated with adherence to referral guidelines in the management of 0–59 days old sick young infants

| Characteristic | Levels | Management according to guideline | | Crude OR (95% CI) | Adjusted OR (95% CI) |
|---|---|---|---|---|---|
| | | Correct (n) | Not correct (n) | | |
| Facility-owned landline or mobile phone | Not available | 145 | 42 | 1.0 | 1.0 |
| | Available | 31 | 10 | 0.90 (0.26 to 3.07) | 1.12 (0.27 to 4.68) |
| Mean score of essential medicine | 0%–33% | 7 | 5 | 1.0 | 1.0 |
| | 34%–66% | 112 | 32 | 2.50 (0.83 to 7.51) | 4.49 (0.93 to 21.77) |
| | 67%–100% | 57 | 15 | 2.71 (0.51 to 14.46) | 12.95 (1.49 to 112.82) |
| No of health extension workers | 1 | 27 | 14 | 1.0 | 1.0 |
| | 2 or more | 149 | 38 | 2.03 (0.37 to 11.28) | 8.51 (0.83 to 86.98) |
| Distance tertiles | 0–8 km | 54 | 23 | 1.0 | 1.0 |
| | 8.01–20 km | 63 | 18 | 1.49 (0.37 to 6.03) | 1.01 (0.17 to 6.13) |
| | 20.01 or more km | 59 | 11 | 2.28 (0.56 to 9.30) | 0.58 (0.08 to 4.20) |
| Age | 0–1 week | 49 | 10 | 1.0 | 1.0 |
| | 2–4 weeks | 69 | 29 | 0.49 (0.15 to 1.58) | 0.57 (0.15 to 2.24) |
| | 5 or more weeks | 58 | 13 | 0.91 (0.26 to 3.23) | 1.37 (0.28 to 6.76) |
| Gender | Boy | 95 | 28 | 1.0 | 1.0 |
| | Girl | 81 | 24 | 0.99 (0.47 to 2.11) | 1.19 (0.55 to 2.58) |

**Table 6** Factors associated with adherence to referral guidelines in the management of 2–59 months old sick children

| Characteristic | Levels | Management according to guideline | | Crude OR (95% CI) | Adjusted OR (95% CI) |
| --- | --- | --- | --- | --- | --- |
| | | Correct (n) | Not correct (n) | | |
| Age | 0–1 week | 282 | 27 | 1.0 | 1.0 |
| | 2–4 weeks | 315 | 26 | 1.16 (0.65 to 2.06) | 1.24 (0.70 to 2.18) |
| | 5 or more weeks | 402 | 22 | 1.75 (0.95 to 3.23) | 1.78 (0.93 to 3.40) |
| Gender | Boy | 524 | 37 | 1.0 | 1.0 |
| | Girl | 475 | 38 | 0.88 (0.56 to 1.39) | 0.97 (0.61 to 1.55) |
| Amoxicillin | Not available | 153 | 26 | 1.0 | 1.0 |
| | Available | 846 | 49 | 2.93 (1.48 to 5.82) | 2.74 (1.40 to 5.35) |
| Distance tertiles | 0–8 km | 340 | 24 | 1.0 | 1.0 |
| | 8.01–14 km | 298 | 28 | 0.75 (0.36 to 1.58) | 0.72 (0.28 to 1.82) |
| | 14.01 or more km | 361 | 23 | 1.11 (0.54 to 2.27) | 0.96 (0.44 to 2.05) |
| Mean score of basic amenities | 0%–33% | 130 | 19 | 1.0 | 1.0 |
| | 34%–66% | 278 | 16 | 2.54 (1.01 to 6.39) | 1.97 (0.71 to 5.49) |
| | 67%–100% | 591 | 40 | 2.16 (0.93 to 5.02) | 1.41 (0.56 to 3.56) |
| Mean score of basic equipment | 0%–33% | 61 | 5 | 1.0 | 1.0 |
| | 34%–66% | 141 | 18 | 0.64 (0.21 to 1.93) | 0.96 (0.12 to 7.57) |
| | 67%–100% | 797 | 52 | 1.26 (0.45 to 3.47) | 2.46 (0.29 to 20.54) |
| Mean score of diagnostics | 0%–33% | 448 | 46 | 1.0 | 1.0 |
| | 67%–100% | 551 | 29 | 1.95 (1.10 to 3.46) | 4.59 (1.79 to 11.75) |
| Mean score of overall readiness | 0%–33% | 21 | 7 | 1.0 | 1.0 |
| | 34%–66% | 570 | 46 | 4.13 (0.68 to 25.02) | 3.19 (0.43 to 23.61) |
| | 67%–100% | 408 | 22 | 6.18 (1.01 to 37.87) | 1.59 (0.14 to 18.00) |
| No of health extension workers | 1 | 193 | 23 | 1.0 | 1.0 |
| | 2 or more | 806 | 52 | 1.84 (0.92 to 3.72) | 1.52 (0.77 to 3.02) |

many care-seeking sick children were not getting the care they required, at least not at the health post.[35] The lack of essential medicines might be a reason for some referrals of children with non-severe diseases, which otherwise may be treated at the first-level health facility.[32 36] This condition might also impose caregiver for unnecessary social and economic costs incurred by referrals of children with non-severe illnesses.[36 37]

Indeed, our data also showed that health posts that had medicine and diagnostics were likely to adhere to referral guidelines. This calls for the need of equipping the health facilities with the basic essential medicines and diagnostics. Although not statistically significant in the multivariable analysis, health posts with two or more health extension works were also more likely to refer as well as adhere to referral guidelines for young infants. A study in Niger also found adequate staffing to be linked to adherence to referral guidelines.[29]

The non-adherence of the some of the healthcare providers to the referral guidelines when managing severely sick children found in this study requires improvement. This problem could result in overuse of antibiotics, and missing treatment or increased lethality in severe cases.[38] Hence, mechanisms that can improve adherence to referral guidelines are needed as part of actions for improved quality of care.

## CONCLUSION

We have shown that health extension workers referred few children with severe illnesses while a referral to the next-level health centre was recommended by guidelines. A significant proportion of the health posts were not ready to provide the necessary care, including referral. The severity of the child's illness, the availability of essential medicines and rapid diagnostic test were among the predictors of referrals of sick children from the health posts. Among other actions to enhance the quality of care provided at the health posts, improving medicine and diagnostic supplies may strengthen adherence to referral guidelines and ultimately prevent child deaths in Ethiopia.

**Author affiliations**
[1]Regional Health Bureau, Southern Nations Nationalities and Peoples' Region, Hawassa, Ethiopia

[2]College of Medicine and Health Sciences, School of Public Health, Hawassa University, Hawassa, Sidama, Ethiopia
[3]College of Health Sciences, Department of Paediatrics and Child Health, Addis Ababa University, Addis Ababa, Ethiopia
[4]London School of Hygiene and Tropical Medicine, Faculty of Infectious and Tropical Diseases, London, UK
[5]Health System and Reproductive Health Research Directorate, Ethiopian Public Health Institute, Addis Ababa, Ethiopia
[6]Epidemiology and Biostatistics, Institute of Public Health, College of Medicine and Health Sciences, University of Gondar, Gondar, Ethiopia

**Correction notice** This article has been corrected since it was first published. The name for the author "Henok Tadele" has been corrected.

**Acknowledgements** We would like to extend our sincere gratitude to SNNP Regional Health Bureau and Hawassa University, College of Medicine and Health Sciences for providing support to conduct this research project. We would also like to pass our deepest appreciation to the different levels of the health systems and health providers involved in this study.

**Contributors** HB contributed to the conceptualisation of the study, design, development of the survey instrument, training and supervision of data collectors, statistical analysis and wrote the original draft and final versions of the manuscript. DHK and HDT contributed to the conceptualisation of the study design, development of the study instruments, supervision of the data collection process, data analysis and review of the different versions of the manuscript. DB and LP contributed to the conceptualisation of the study, design, development of the survey instrument, data compilation and statistical analysis, and reviewed the different versions of the paper. AD contributed to the management of distance from health post to health centre data, development of the study instruments, supervision of the data collection process, and review of the different versions of the manuscript. All authors read and approved the final version of the manuscript.

**Funding** The study was supported by a grant from Bill and Melinda Gates Foundation to The London School of Hygiene and Tropical Medicine (grant OPP1132551).

**Disclaimer** The funder had no role in data collection, analysis, or interpretation of results.

**Competing interests** None declared.

**Patient consent for publication** Not required.

**Ethics approval** Ethical approvals were secured from the ethical review boards of Hawassa University (protocol number IRB/199/10; June 2018), the Ethiopian Public Health Institute (protocol number SERO-012-8-2016; V.001, August 2016) and the London School of Hygiene & Tropical Medicine (protocol number 11235; June 2016).

**Provenance and peer review** Not commissioned; externally peer reviewed.

**Data availability statement** Data are available on reasonable request. Data may be obtained from a third party and are not publicly available. All data relevant to the study are included in the article or uploaded as online supplemental information. Data from this study are co-owned by the participating institutions and stored in a depository at the Ethiopian Public Health Institute (EPHI). The use of these data is guided by a data sharing agreement. Data can be accessed from the secretary of the data sharing committee of EPHI-LSHTM collaborative projects. Contact information: Martha Zeweldemariam, email: martha.zeweldemariam@lshtm.ac.uk.

**ORCID iDs**
Habtamu Beyene http://orcid.org/0000-0003-0923-5191
Lars Persson http://orcid.org/0000-0003-0710-7954
Atkure Defar http://orcid.org/0000-0001-9435-2135
Della Berhanu http://orcid.org/0000-0002-4984-893X

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
