## [Reviewer comments · BMJ Open]

ARTICLE DETAILS

TITLE (PROVISIONAL)	Factors associated with the referral of children with severe illnesses at primary care level in Ethiopia: a cross-sectional study.
AUTHORS	Beyene, Habtamu; Kassa, Dejene; Tadele, Henok; Persson, Lars; Defar, Atkure; Berhanu, Della

VERSION 1 – REVIEW

REVIEWER	Tedbabe Degefie Newborn and Child Health Consultant, UNICEF
REVIEW RETURNED	21-Jan-2021

GENERAL COMMENTS	The paper is well written addressing important newborn and child health issues. It will be useful to mention the possibility of incomplete documentation as one source of bias
--

REVIEWER	Phillip Wanduru Makerere University
REVIEW RETURNED	25-Jan-2021

GENERAL COMMENTS	Thank you for the opportunity to review this article. I find that it is an important paper with important findings. My main comment is that authors should take time to align the study aim with the methods, results, and then key findings. This is so important for readers to fully comprehend the important information being published. The study aim is: "identify factors associated with referral..". However, in the results we have factors associated with referral; and also, factors associated with correct management, and correct or wrong referral. While I agree that all these results are important, they should be included in the study objectives and methods so that readers follow well. The authors also need to fully describe the way logistic regression was done. Some of the tables can be combined for-example; tables 3, 4, and 5 can be combined into one or two tables. There can be a table on factors for those who are 0-59 days, and another table for 2-59 days. The discussion section: The authors need to discuss their key findings and explain their implications to practice. The authors also put "results" in the "discussion section". This is on page 14 lines 25-50. On page 13, line 17; they mention "insufficient proportion of infants who required referral were referred". This language can be improved to maybe "less than half...". I believe this is an important finding
--

	from this study.
REVIEWER	Helena Hildenwall Karolinska Institutet, Global Public Health
REVIEW RETURNED	29-Jan-2021

GENERAL COMMENTS	Dear Authors this is indeed a topic of great importance and better understanding of referral practises and how these can be refined are essential for the achievement of the Sustainable Development Goal 3.2. Unfortunately, I think the manuscript suffer from some important methodological issues that have not been sufficiently discussed and clarified. MAJOR COMMENTS  1. the use of register data to classify children and the impact on what you have really assessed - is it the referral practises or rather the quality of documentation? I have not done any research in Ethiopia but in other low income settings and in my experience the reliability of facility records is poor. As an example, sepsis is a common diagnosis in primary care facilities in Malawi and the vast majority of children with that diagnosis will be sent home. My point here is that what you find in the registers may be more a reflection of poor documentation rather than clinical limitation. The fact that the clinician may have assessed the child correctly but documented it poorly must be discussed. 2. It seems data collection involved one single visit to the included facilities though the registers covered the past months. The availability of medicines may well have varied a lot during that time. How is that taken into account? 3. You report that some of the facilities have not at all managed children during the past three months? How is that possible? 4. You explain well the use of the different domains in the WHO health service availability and readiness assessment tool but then also report on the role of availability of rdts and amoxicillin The reason for chosing these two items specifically are not explained and thus it may seem that some data fishing could have occurred. 5. The tables 3+4 are not easy to understand and maybe not needed?! Or in need of clarification 6. a lot of the information under "factors associated with referrals" are repetition of what is said in the previous paragraph "referrals in different groups" MINOR mix of past and present tense - pls revise all text that also may need some overall language editing you report that health posts in the lowest and highest tertiles of medicine availability referred the fewest children. Presumably you refer to proportions? Please clarify?
--

VERSION 1 – AUTHOR RESPONSE

Reviewer: 1 Dr. Tedbabe Degeffie, Newborn and Child Health Consultant

Comments to the Author:

The paper is well written addressing important newborn and child health issues. It will be useful to mention the possibility of incomplete documentation as one source of bias.

Response: Thank you. We have incorporated incomplete documentation as a possible source of bias on page 3 in the 4th bullet, and on page 14, 1st and 2nd lines in the main document.

Reviewer: 2 Mr. Phillip Wanduru, Makerere University

Comments to the Author:

Thank you for the opportunity to review this article. I find that it is an important paper with important findings.

1. My main comment is that authors should take time to align the study aim with the methods, results, and then key findings. This is so important for readers to fully comprehend the important information being published.

The study aim is: "identify factors associated with referral...". However, in the results we have factors associated with referral; and also, factors associated with correct management, and correct or wrong referral. While I agree that all these results are important, they should be included in the study objectives and methods so that readers follow well.

Response: Thank you. We have modified the objectives in the abstract (page 2 line 3) and introduction (page 5 line 8 and 9) in such a way that the the analysis of healthcare providers' adherence to referral guidelines is part of the objectives.

2. The authors also need to fully describe the way logistic regression was done.

Response: Thank you. We describe the multiple logistic regression analysis in the last paragraph of page 8 and the 1st paragraph of page 9. We mention that relevant factors with a statistically significant association to the referral of sick children in the bivariate analyses were entered into the multivariable logistic regression analysis to identify referral predictors. The same approach was used for healthcare providers' adherence to referral guidelines (page 9, 1st paragraph).

3. Some of the tables can be combined for-example; tables 3, 4, and 5 can be combined into one or two tables. There can be a table on factors for those who are 0-59 days, and another table for 2-59 days.

Response: Thank you. We have combined table 4 and 5 (now table 3 on page 33). We prefer to move table 3 to become supplementary table 1 (page 35); to be accessible for readers interested in detailed information.

4. The discussion section: The authors need to discuss their key findings and explain their implications to practice. The authors also put "results" in the "discussion section". This is on page 14 lines 25-50.

Response: Thank you. We have modified the discussion part. We have tried to avoid repeating results in the discussion (except in the first paragraph, which summarizes the main findings of the study). We have tried to emphasize the public health implications of the findings (Page 14 - 15).

5. On page 13, line 17; they mention "insufficient proportion of infants who required referral were referred". This language can be improved to maybe "less than half....". I believe this is an important finding from this study.

Response: Thank you. We have modified the sentence accordingly on page 14 2nd paragraph line 2.

Reviewer: 3 Dr. Helena Hildenwall, Karolinska Institutet

Comments to the Author:

Dear Authors

This is indeed a topic of great importance and better understanding of referral practises and how these can be refined are essential for the achievement of the Sustainable Development Goal 3.2. Unfortunately, I think the manuscript suffer from some important methodological issues that have not been sufficiently discussed and clarified.

MAJOR COMMENTS

1. The use of register data to classify children and the impact on what you have really assessed - is it the referral practises or rather the quality of documentation? I have not done any research in Ethiopia but in other low income settings and in my experience the reliability of facility records is poor. As an example, sepsis is a common diagnosis in primary care facilities in Malawi and the vast majority of children with that diagnosis will be sent home. My point here is that what you find in the registers may be more a reflection of poor documentation rather than clinical limitation. The fact that the clinician may have assessed the child correctly but documented it poorly must be discussed.

Response: We appreciate your comment. Poor documentation in some health facilities is likely. On page 3 in the 4th bullet, and page14, lines 4 - 10, we have added that poor and incomplete documentation in registers may have occurred. However, we would like to indicate that the registers at Ethiopian health facilities are not just a recording and reporting tool, but also serve as a job aid, providing the algorithm ASSESS, CLASSIFY AND TREAT. The algorithm follows the community-based newborn care and integrated community-case management guidelines. Hence, despite the possibility of data quality problem, we believe that these care and treatment registers could still reflect the existing clinical practices among the health extension workers at the health post. The referral adherence trends shown in our study (through register reviews) were in line with studies, which observed the clinical competency and adherence to guidelines of the HEWs in the same study areas (Getachew et al., 2019).

2. It seems data collection involved one single visit to the included facilities though the registers covered the past months. The availability of medicines may well have varied a lot during that time. How is that taken into account?

Response: Thank you. This is also important concern, although it applies less for the 2-59 months old children as we included the most recent ten children who were seen closer to the time of the survey. It is, however, relevant for the sick young infants for whom data were collected for the three months preceding the survey. We have noted this as a limitation in the study indicating the need to interpret the association between drug availability and referral with caution on page 13 in the last paragraph and 1st line of page 14.

3. You report that some of the facilities have not at all managed children during the past three months? How is that possible?

Response: Thank you. There were 117 health posts, which did not give child health services to any 0-59 day old young infant in the three months preceding the survey and 18 health posts without 2-59 months old child in the three months preceding the survey. This low level of services for under-five children was the rationale for the Optimization of the Health Extension Program intervention. This study is a part of the evaluation of that intervention.

4. You explain well the use of the different domains in the WHO health service availability and readiness assessment tool but then also report on the role of availability of rdts and amoxicillin The reason for choosing these two items specifically are not explained and thus it may seem that some data fishing could have occurred.

Response: Thank you. We specifically explained about amoxicillin and RDTs, due the significant role of these essential commodities in treating presumed severe bacterial infection (or very severe disease) and for the diagnosis of malaria, respectively.

5. The tables 3+4 are not easy to understand and maybe not needed?! Or in need of clarification

Response: Thank you. Instead of totally removing table 3 and 4 from the manuscript, we have moved table 3 to become supplementary table 1 (page 35) – for readers interested in details. We have merged table 4 and 5 so that those data are now found in table 3, on page 33.

6. A lot of the information under "factors associated with referrals" are repetition of what is said in the previous paragraph "referrals in different groups"

Response: Thank you. We have now integrated the two sub-headings and removed redundant information. (Page 10 - 11).

MINOR

Mix of past and present tense - pls revise all text that also may need some overall language editing

Response: Thank you. We have revised the text and made changes accordingly.

You report that health posts in the lowest and highest tertiles of medicine availability referred the fewest children. Presumably you refer to proportions? Please clarify?

Response: Thank you. This statement is already modified while integrating the two paragraphs into one, page 11.

VERSION 2 – REVIEW

REVIEWER	Phillip Wanduru Makerere University
REVIEW RETURNED	29-Apr-2021

GENERAL COMMENTS	I think that the paper has been improved significantly from the last version. It is now much easier comprehend. I would like state that this is quite an important paper for referral in the context of CHWs. The key message that seems to come out is that diagnostic capacity, and presence of medicines (anti-biotics and other amenities) is likely to improve the way referral is done. My main comment:  1. The authors need to clarify: is the problem with “all referrals”, or “incorrect (those done without adhering to standards) referrals”? In my opinion conducting an analysis that assesses factors associated with “non-adherence to referral guidelines” would be more useful. These results are in fact presented in supplementary tables 4 and 5. The current table 3 in the results mixes all referrals (the correct ones and wrong ones). The authors could consider having supplementary tables 4 and 5 moved to results. Also, to use the phrase “factors associated with incorrect adherence to referral guidelines”. 2. I also find that table 3 in the supplementary tables is quite important, the authors may consider moving it to the results. Minor comments:  1. I find the table 2, quite informative. For example, only 36% of 0–59-day babies with possible serious bacterial infection get referred. Only 26% of those 2-59months; with severe malnutrition get
--

	referred. The authors could say more about these results. 2. Correct the spelling of “months” in table 2, page 25. 3. Some conditions that are characterized as requiring (or not) referral, but for me these SOPs can be debatable. For example, pneumonia, the authors indicate that it should not be referred. I think that is dependent on the severity. May be use the word “cough” but a serious pneumonia may require referral.
--	---

VERSION 2 – AUTHOR RESPONSE

Reviewer: 2

Mr. Phillip Wanduru, Makerere University

Comments to the Author:

I think that the paper has been improved significantly from the last version. It is now much easier comprehend.

I would like state that this is quite an important paper for referral in the context of CHWs. The key message that seems to come out is that diagnostic capacity, and presence of medicines (anti-biotics and other amenities) is likely to improve the way referral is done.

My main comment:

1. The authors need to clarify: is the problem with “all referrals”, or “incorrect (those done without adhering to standards) referrals”? In my opinion conducting an analysis that assesses factors associated with “non-adherence to referral guidelines” would be more useful. These results are in fact presented in supplementary tables 4 and 5. The current table 3 in the results mixes all referrals (the correct ones and wrong ones). The authors could consider having supplementary tables 4 and 5 moved to results. Also, to use the phrase “factors associated with incorrect adherence to referral guidelines”.

Response: Thank you for an important suggestion. Adherence to guidelines was one aspect of this study. As you can see in the methods section on page 9 of 1st paragraph, from lines 2 – 4, we analysed adherence to referral guidelines, both “correctly” and “Incorrectly” managed. As suggested, we have moved previous supplementary tables 4 and 5 to the result section, now labelled table 5 and 6, at page 30 and 31, respectively.

2. I also find that table 3 in the supplementary tables is quite important, the authors may consider moving it to the results.

Response: Thank you. Supplementary table 3 is now moved to the result section, labelled Table 4 on page 29.

Minor comments:

1. I find the table 2, quite informative. For example, only 36% of 0–59-day babies with possible serious bacterial infection get referred. Only 26% of those 2-59months; with severe malnutrition get referred. The authors could say more about these results.

Response: Thank you. We have added comments to these findings (Page 10, 2nd paragraph, lines 5 – 7).

2. Correct the spelling of “months” in table 2, page 25.

Response: Thank you. Done. Now, it is on page 24.

3. Some conditions that are characterized as requiring (or not) referral, but for me these SOPs can be debatable. For example, pneumonia, the authors indicate that it should not be referred. I think that is dependent on the severity. May be use the word “cough” but a serious pneumonia may require referral.

Response: Thank you for your comment on pneumonia. As you can see from Table 2, the ICCM and IMNCI guidelines in Ethiopia have the following classifications: no pneumonia, pneumonia, or severe pneumonia. In Table 2, children with severe pneumonia and pneumonia are shown separately. Those with cough but no pneumonia are already included with others as shown in the footnote.